# Physical Activity and Psychonutritional Correlates of Eating Disorder Risk in Female Health Science Students

**DOI:** 10.3390/healthcare13141679

**Published:** 2025-07-11

**Authors:** Patricia Ruiz-Bravo, Germán Díaz Ureña, Bárbara Rodríguez-Rodríguez, Nuria Mendoza Laiz, Sonia García-Merino

**Affiliations:** Faculty of Health Sciences, Universidad Francisco de Vitoria, 28223 Madrid, Spain; p.ruiz.prof@ufv.es (P.R.-B.); br.rodriguez.prof@ufv.es (B.R.-R.); nuria.mendoza@ufv.es (N.M.L.); sonia.garciamerino@ufv.es (S.G.-M.)

**Keywords:** body image, self-concept, exercise, diet, Mediterranean, students, health occupations

## Abstract

**Objective:** This study sought to examine the correlation between physical activity levels and various psychological and nutritional factors associated with the risk of developing eating disorders among female university students in the Health Sciences discipline. **Method:** The study assessed body image, self-esteem, nutritional status, adherence to the Mediterranean diet, and attitudes toward food in a sample of 96 women, categorized into two groups based on their level of physical activity. **Results:** Significant differences in skeletal muscle mass were identified between the groups, alongside associations between body dissatisfaction, low self-esteem, and elevated EAT-26 scores. Furthermore, students with higher levels of physical activity exhibited a significantly increased prevalence of eating disorder risk. Logistic regression analysis identified body dissatisfaction as a significant predictor of eating disorder risk, while membership in the group with the highest physical activity levels enhanced model fit and increased eating disorder risk. **Conclusions:** These findings indicate that, while physical activity is associated with certain benefits related to body composition and eating habits, it may also be linked to a heightened risk of disordered eating behaviors, contingent upon the underlying motivations and body perceptions involved. This study highlights the necessity for comprehensive preventive strategies that address both the physical and psychological dimensions of physical activity in female university students.

## 1. Introduction

Eating disorders (EDs) represent a substantial public health issue, particularly prevalent among adolescents and young adults, with a higher incidence observed in females [1]. These disorders are defined by persistent disruptions in eating or eating-related behaviors that adversely impact physical health and psychosocial functioning [2]. The risk of developing EDs is influenced by a complex interplay of psychological, behavioral, and nutritional factors, among which physical activity (PA), body image perception, self-esteem, nutritional status, and adherence to healthy dietary patterns, such as the Mediterranean diet, are particularly significant [3,4,5,6].

PA is evidenced to have a dual impact on diet-related health outcomes. Regular participation in moderate-to-vigorous physical activity is correlated with healthier eating attitudes, a reduced preference for ultra-processed foods, and a stronger adherence to balanced dietary patterns [3,4,7]. Conversely, high levels of physical activity, particularly when motivated by concerns related to weight control or body image, may be linked to disordered eating behaviors, indicating a complex relationship between the benefits and potential risks of exercise in this context [8]. The intensity and motivation for engaging in PA appear to be key determinants of this relationship [4].

This nuanced relationship is also reflected in findings from specific physical activity contexts, where the type and focus of the sport appear to influence body image and eating behaviors differently. The risk of developing ED psychopathology among female athletes varies notably, depending on the type of sport. According to findings from this meta-analysis, athletes involved in aesthetic or leanness-focused sports—such as gymnastics, figure skating, dance, or long-distance running—report higher levels of body dissatisfaction, drive for thinness, dietary restraint, and loss-of-control eating compared to those in non-aesthetic or non-leanness-focused sports, such as team sports. These differences suggest that the body ideals emphasized in certain disciplines may heighten vulnerability to disordered eating behaviors. In contrast, athletes in non-aesthetic sports reported significantly lower levels of body dissatisfaction and drive for thinness than non-athletes, indicating that participation in these sports may offer some protective benefits against body image concerns [9].

Furthermore, the perception of body image is a critical determinant in the emergence of dysfunctional eating behaviors. Body dissatisfaction and low self-esteem have been consistently linked to an increased likelihood of developing EDs [1,5,10,11]. This dissatisfaction, which may occur even among individuals with a body mass index (BMI) within the normal range, often persists from adolescence into adulthood. This relationship is shaped by factors such as social pressure, the internalization of aesthetic ideals, and experiences of body stigmatization [1]. Compared to men, women generally exhibit greater body dissatisfaction and are more susceptible to idealized body standards, thereby heightening their risk of developing disordered eating behaviors [12,13,14].

In this regard, nutritional status and BMI also interact with these factors. However, subjective weight perception appears to be a more accurate predictor of eating attitudes than objective BMI measurements. Young individuals with a higher BMI or those who perceive themselves as overweight tend to exhibit poorer relationships with food and experience greater body dissatisfaction [15,16,17]. Concurrently, adherence to the Mediterranean diet (MEDAS)—recognized for its protective effect against multiple chronic diseases—has been shown to be low among university students. Nevertheless, it is positively associated with lower BMI, improved diet quality, and enhanced academic performance [18,19]. Furthermore, recent studies have indicated that MEDAS is associated with reduced body dissatisfaction and a decreased risk of developing EDs [20,21].

Likewise, regular PA is associated with enhanced body image perception and increased MEDAS, which together may function as protective factors against EDs [22].

Moreover, evidence from adolescent and university populations indicates that higher adherence to the Mediterranean diet, when combined with regular physical activity, is positively associated with self-esteem, emotional well-being, and improved body composition [23]. These findings support the relevance of examining the interaction between sport participation and dietary patterns, particularly in populations vulnerable to body dissatisfaction and disordered eating [24].

Self-esteem exerts a direct influence on body image and EDs. Studies have indicated that low self-esteem is a significant risk factor for the development of EDs, as well as for increased body-related concerns [25,26]. Furthermore, when perfectionism is coupled with low self-esteem, it serves as a significant predictor of ED symptomatology [27,28]. However, some studies suggest a non-causal relationship between self-esteem and EDs, highlighting the complexity of this construct [29].

In view of these considerations, the primary aim of this study was to analyze the relationship between PA level and various psychological and nutritional variables associated with the risk of developing EDs in female university students in the Health Sciences discipline. Specifically, this study sought to examine the differences between groups with higher and lower PA levels in terms of body shape concern, self-esteem, nutritional status, and MEDAS. Additionally, we explored whether a high level of PA was associated with a greater prevalence of EDs.

## 2. Materials and Methods

### 2.1. Study Design

This study employed a cross-sectional, descriptive observational design. Participant recruitment was conducted through quota sampling to ensure the representation of students from all four degree programs within the Faculty of Health Sciences. Data collection was obtained through self-administered questionnaires distributed in a digital format using Microsoft Forms (version 365) and disseminated via an institutional digital platform (CANVAS). The STROBE guidelines for observational studies in epidemiology were followed [30].

### 2.2. Participants

The study’s sample comprised female university students. Prospective participants were informed about the study through announcements disseminated via email. An online registration form was provided for enrollment. The sample composition ensured a proportional representation of students from each degree program within the overall female student population of the faculty. This proportional sampling approach introduced a certain degree of heterogeneity among participants, as students from diverse academic contexts were included; however, this was essential to preserve the representativeness of the sample and to reflect the actual distribution of the student population across different degree programs.

The inclusion criteria were as follows: being female, enrolled in one of the degree programs within the Faculty of Health Sciences (Sport Sciences, Nutrition, Nursing, or Kinesiology), and aged between 18 and 25 years. Participants who did not complete all the required assessments and/or did not provide informed consent were excluded from the study.

An initial sample size of 92 participants was estimated based on the existing scientific literature, which indicates an ED prevalence of 20% in the Spanish athletic population and 4% in the general population [31]. With a significance level of α = 0.05, a statistical power of 1 − β = 0.8, and a group ratio of 1, a target distribution was established based on the proportion of students in each degree program: Sport Sciences (*n* = 9), Nursing (*n* = 52), Kinesiology (*n* = 20), and Nutrition (*n* = 11). Anticipating a 10% dropout rate and to ensure that the estimated sample size was met, the recruitment target was set at 101 participants.

### 2.3. Procedure

The questionnaires were distributed across four consecutive weeks to minimize participant fatigue and reduce the risk of response bias associated with completing multiple psychometric instruments in a single session. Each week, participants received one or two questionnaires, allowing them to respond with better attention and accuracy. During “Week 1”, the International Physical Activity Questionnaire (IPAQ) was administered to assess the participants’ PA levels. In “Week 2”, participants completed nutrition-related questionnaires ((Eating Attitudes Test (EAT-26) and Prevention with Mediterranean Diet (PREDIMED)). During “Week 3”, questionnaires related to body dissatisfaction, the BSQ, and the Rosenberg Self-Esteem Scale (RSE) were completed. Finally, in “Week 4”, body composition was assessed using the InBody model 770 device to determine nutritional status.

### 2.4. Variables

The variables examined in this study were as follows:Physical activity (PA);Mediterranean diet adherence (MEDAS);Eating disorders (EDs);Body shape concern (BSC);Self-esteem;Nutritional status: weight, body mass index (BMI), skeletal muscle mass (SMM), percent body fat (PBF), body fat mass (BFM), and visceral fat area (VFA).

### 2.5. Measures

The following instruments were employed in this study.

#### 2.5.1. Instrument for Assessing PA

PA was assessed using the short form of the IPAQ [32]. This instrument comprises seven items and yields data on the duration of walking, engagement in moderate- and vigorous-intensity activities, and sedentary behaviors over the preceding seven days. It encompasses four domains of physical activity: leisure time, domestic and household activities, occupational activity, and transportation. The questionnaire is recommended for use in adults aged 18–69 years [33]. For interpretation, all metabolic expenditures were aggregated and expressed in METs per week. Vigorous, moderate, and light activities were multiplied by 8, 4, and 3.3 METs, respectively. Following the authors’ recommendations for this questionnaire, an initial screening was conducted to ensure that no participant reported more than 16 h of daily activity. Additionally, the time reported for each PA was capped at a maximum of 4 h, that is, any value exceeding 4 h was replaced with 4 h.

#### 2.5.2. Instruments for Assessing MEDAS

The MEDAS was assessed using the PREDIMED questionnaire [34]. This questionnaire comprises 14 items that assess the number of servings and frequency of consumption of typical Mediterranean foods or food groups, including olive oil, nuts, fruits, wine, seafood, and legumes. Additionally, it includes questions regarding the limited consumption of foods not traditionally associated with the Mediterranean diet, such as red or processed meats, sugary beverages, desserts, and sweets. Each point scored corresponds to a greater adherence to the Mediterranean diet. Scores approaching 14 indicate a high level of adherence; scores between 8 and 11 suggest moderate adherence; scores between 5 and 7 denote low adherence; and scores of 5 or below reflect very low adherence.

#### 2.5.3. Instrument for Assessing Attitudes Toward Food

The assessment of attitudes toward food was conducted utilizing the EAT-26 instrument, which is specifically designed to identify symptoms and concerns associated with the fear of weight gain, the drive for thinness, and the presence of restrictive eating behavior patterns.

The questionnaire consists of 26 items [35], each evaluated on a six-point Likert scale. The total score is obtained by recoding the responses as follows: scores from 1 to 3 are coded as 0, a score of 4 is recoded as 1, a score of 5 as 2, and a score of 6 as 3. The only exception is item 25, which is reverse-scored. The threshold for differentiating between asymptomatic and symptomatic individuals is established at 19 points [35]. Accordingly, individuals with scores below 20 were classified as not at risk for EDs, whereas those with scores equal to or greater than 20 were classified as being at high risk for EDs.

#### 2.5.4. Instrument Used to Assess BSC

BSC was assessed using the BSQ-18. This questionnaire allows for the evaluation of body satisfaction levels and the identification of dissatisfaction related to body weight [36]. The questionnaire consists of 18 items in which participants are asked to indicate their level of satisfaction with their body image over the past four weeks. Responses are given on a six-point Likert scale (Never = 1 to Always = 6).

#### 2.5.5. Instrument Used to Assess Self-Esteem

Self-esteem was evaluated utilizing the Rosenberg Self-Esteem Scale (RSE), a questionnaire developed by Rosenberg to measure self-esteem based on a Guttman scale. In this study, the Spanish-translated version was used [37]. Moreover, it has been validated within university student populations [38]. It is a 10-item scale that assesses self-respect and self-acceptance, rated on a 4-point Likert scale ranging from 1 (strongly disagree) to 4 (strongly agree). Low self-esteem responses are indicated by ‘disagree’ or ‘strongly disagree’ on positively worded items (1, 3, 4, 7, and 10) and by ‘strongly agree’ or ‘agree’ on negatively worded items (2, 5, 6, 8, and 9). High self-esteem is interpreted for those scoring 30 points or above, medium self-esteem corresponds to total scores between 20 and 27, and low self-esteem is defined as scores of 26 points or below.

#### 2.5.6. Assessment of Nutritional Status

Nutritional status was assessed using anthropometric indicators (e.g., weight, height, BMI, body fat percentage, visceral fat area), in accordance with the “A” (Anthropometric) component of the ABCD model of nutritional assessment (Anthropometric, Biochemical, Clinical, and Dietary) [39].

The InBody 770 device (Seoul, Republic of Korea) was employed to evaluate nutritional status. The parameters obtained included weight, BMI, SMM, PBF, BFM, and VFA. The protocol followed was as follows.

First, participant information—including sex, age, and height—was recorded on the screens of the three devices and in Microsoft Excel (version 365).

To measure participants’ height, a stadiometer (SECA 700) was employed in accordance with the International Society for the Advancement of Kinanthropometry (ISAK) protocol [40]. Participants were asked to remove their shoes and take a deep breath, holding it during the measurement. Weight and body composition data were obtained using the InBody 770 scale through direct segmental multi-frequency bioelectrical impedance analysis, after cleaning the device with alcohol. Measurement requirements included abstaining from moderate-to-intense physical activity within the previous 24 h, urinating at least 30 min prior, being barefoot, not wearing metallic accessories, and fasting for at least four hours. Compliance with these requirements was confirmed before the measurement. The device is equipped with tetrapolar tactile electrodes with eight contact points—two on each hand and foot—which perform impedance measurements at six different frequencies (1, 5, 50, 250, 500, and 1000 kHz) for each body segment. The skin on the soles of the feet and palms of the hands were cleaned with an electroconductive wipe before participants positioned themselves on the platform’s tactile electrodes, maintaining an upright posture while holding the handles with approximately 20° of lateral arm abduction and about 30° of scapulohumeral joint flexion, ensuring contact with all eight electrode points. It was verified that the legs and thighs, as well as the arms and torso, were not in contact before activating the device, which sends an electrical current through the participant’s body to estimate various indicators of body composition [41]. All information was recorded in Microsoft Excel (version 365), as previously mentioned.

### 2.6. Data Analysis

To distinguish participants with higher PA levels from those with lower levels, the sample was divided into two groups based on the median level of PA.

Descriptive statistical analyses were conducted for all variables, including measures of central tendency and dispersion: mean, median, standard deviation, and interquartile range. Pearson’s correlation coefficient was used to assess correlations between variables, with Holm’s adjustment applied for multiple comparisons.

Mean comparisons between the two independent groups were performed using Welch’s *t*-test. Effect sizes for mean differences were calculated using Cohen’s d and interpreted following Cohen (1988) [42] (small: 0.2, medium: 0.5, large: 0.8).

A Chi-square test with Yates’ continuity correction and Fisher’s exact test were used to evaluate the association between physical activity group and the presence of EDs. Epidemiological analysis was supplemented with prevalence estimates and prevalence ratios to determine the relative risk of EDs based on PA level.

To assess the reliability and validity of the EAT-26 questionnaire, Cronbach’s alpha, McDonald’s omega, Kaiser–Meyer–Olkin (KMO), and Bartlett’s test of sphericity were calculated.

A logistic regression model was fitted to examine the association between physical activity group and body dissatisfaction with the risk of EDs, defined as an EAT-26 score ≥ 20. The dependent variable was dichotomized into “at risk” vs. “not at risk,” and the independent variables included physical activity group (Group 1 vs. Group 2) and total score on the BSQ.

Model selection was based on the Akaike Information Criterion (AIC) [43], with the final model chosen as the one with the lowest AIC value. The model was fitted using the glm() function in R, with a binomial distribution and logit link.

Internal validation was conducted using bootstrap resampling (1000 iterations) to assess the stability of the model and the robustness of its estimates. For each resample, model coefficients were recalculated along with key performance metrics: area under the ROC curve (AUC), mean absolute error (MAE), and mean squared error (MSE). Bootstrap distributions were used to derive 95% confidence intervals (2.5th and 97.5th percentiles) for both coefficients and performance metrics. AUC values and their confidence intervals were computed with the pROC package, while MAE and MSE were derived from predicted probabilities and observed outcomes in each resample.

To evaluate the assumptions of logistic regression, several diagnostic checks were conducted. Multicollinearity was assessed using the Variance Inflation Factor (VIF). Model calibration was further examined using the Hosmer–Lemeshow C and H goodness-of-fit statistics, and through a bootstrap-based calibration curve generated with the rms package [44] (1000 resamples). Residual analysis included the inspection of Pearson and deviance residuals to detect potential outliers or model misfit. Additionally, Cook’s distance was computed to identify influential observations; cases exceeding the 4/n threshold were recorded.

All statistical analyses were conducted using RStudio (Version 2024.09.1+394). The packages ggstatsplot [45], epiR [46], Psych [47], pROC [48], and MASS [49] were used to perform statistical tests and generate figures. The level of significance was set at *p* < 0.05.

## 3. Results

### 3.1. Sample

Of the 108 female participants, 96 met the inclusion criteria (Sport Sciences, *n* = 9; Nursing, *n* = 54; Kinesiology, *n* = 22; Nutrition, *n* = 11). The cutoff point used to divide the sample into two groups based on their level of physical activity was 2373 METs/week. Group 1 included participants with more than 2373 METs/week, while Group 2 included those with 2373 METs/week or fewer.

### 3.2. Study Variables

The EAT-26 questionnaire demonstrated reliability and validity values as follows: Cronbach’s alpha = 0.82; McDonald’s omega = 0.88; and a KMO value of 0.69. Bartlett’s test yielded a *p*-value < 0.01 for our sample.

Table 1 presents the descriptive statistics, including the mean and standard deviation (in parentheses) for all study variables.

Figure 1 presents the correlation values among the different variables. Significant correlations (*p* < 0.05) are displayed in bold and with larger font size. A significant positive correlation was observed between body shape concern and EDs; a significant negative correlation between body shape concern and self-esteem; and a significant positive correlation between BSC and nutritional status indicators (weight, BMI, BFM, PBF, and VFA). Additionally, a statistically significant correlation was observed between SMM and PA.

Figure 2 displays the group comparison (*t*-value), its statistical significance (*p*-value), effect size (Cohen’s d), confidence interval (95% CI), and the number of observations related to nutritional status. The body of the figure includes measures of central tendency and dispersion.

There are significant differences in SMM between the two groups, with Group 1 (the higher PA group) showing higher values.

Figure 3 presents measures of central tendency, distribution, and group comparisons for the variables analyzed.

Although no statistically significant differences were observed between the groups for any of the variables, both EDs and MEDAS showed *p*-values close to 0.05, effect sizes approaching a medium magnitude, and confidence intervals with one bound very close to zero. In both cases, participants who engaged in more PA scored higher.

### 3.3. Analysis of the Association Between PA Group and EDs

Statistical tests were conducted to assess the association between PA group and the risk of developing EDs (χ^2^ = 4.4138, *p* = 0.036). These results indicate a significant association between PA group and the risk of EDs. To confirm these findings, Fisher’s exact test was also performed (*p* = 0.03), which supported the presence of this association.

Table 2 presents the prevalence rates of No Risk and At Risk for EDs across both PA groups.

The prevalence ratio for the risk of EDs between the two groups is 8, with Group 1 showing a prevalence of 16.67%, while Group 2 shows a prevalence of 2.08%.

After evaluating multiple models, a logistic regression model was fitted to assess the association between PA group and body shape concern with the presence of high risk of EDs. The dependent variable was the presence of high ED risk (EAT-26 ≥ 20). Independent variables included PA group (Group 1 vs. Group 2) and body shape concern. Group 1 (more active participants) showed a higher risk compared to Group 2 (Table 3 and Figure 4)

Although the PA group did not reach statistical significance, it improved overall model fit. The AIC was 47.161, suggesting acceptable model adequacy.

To account for potential heterogeneity arising from the inclusion of students from different academic programs, we tested “degree program” (DEGREE) as a covariate in the logistic regression model. Although physical activity levels differed significantly across programs, DEGREE was not a significant predictor of eating disorder risk (*p* = 0.18), and its inclusion did not substantially improve model fit (AIC decreased only marginally from 47.161 to 47.137). For reasons of parsimony and interpretability, DEGREE was excluded from the final model.

The logistic regression model exhibited strong discriminatory capacity, with a bootstrapped AUC of 0.87 (95% CI: 0.76–0.97). Regarding calibration, the bootstrapped curve (B = 1000) (Figure 4) showed low error rates—mean absolute error (MAE) of 0.029 and mean squared error (MSE) of 0.00162—with 90% of absolute errors below 0.067, indicating satisfactory predictive accuracy.

As for the model coefficients, the BSQ score consistently emerged as a significant and stable predictor, with a bootstrapped odds ratio ranging from 1.04 to 1.21, suggesting that greater body shape concern was reliably associated with increased ED risk. In contrast, the physical activity (PA) group showed high variability, with a bootstrapped odds ratio ranging from 0.00 to 1.03, reflecting uncertainty about its independent effect.

Model validity was further assessed through diagnostic tests. Multicollinearity was negligible (VIF = 1.00 for all predictors). The Hosmer–Lemeshow C and H statistics confirmed acceptable calibration (C: χ^2^ = 8.70, df = 8, *p* = 0.369; H: χ^2^ = 13.29, df = 8, *p* = 0.102).

Residual analysis revealed no concerning values or patterns (Pearson residuals: −1.21 to 5.42; deviance residuals: −1.34 to 2.61). Cook’s distance reached a maximum of 0.35, with only four observations (cases 66, 88, 92, and 96) exceeding the 4/n threshold (0.0417), but none approaching the critical value of 1. Overall, these results support the model’s robustness and good fit.

## 4. Discussion

The present study aimed to analyze how various psychonutritional factors are related to the level of PA among female university students in health-related fields, with a focus on variables associated with the risk of developing EDs.

In this regard, the data revealed a direct and significant association between body image perception and variables such as eating attitudes and nutritional status. Students who perceived themselves as having a negative body image exhibited higher levels of dissatisfaction and lower self-esteem, which may lead to restrictive eating behaviors as a strategy to control weight and appearance. Furthermore, a significant relationship was found between anthropometric parameters (weight and PBF) and body shape concern, as well as an inverse relationship with self-esteem. Specifically, participants with higher weight and PBF perceived their body image more negatively, which was associated with a greater risk of developing EDs.

This pattern was confirmed in subsequent analyses, where body shape concern was significantly associated with higher scores on the EAT-26, suggesting an increased risk of EDs. Logistic regression further identified body shape concern as a significant predictor of ED risk (coefficient = 0.075, *p* = 0.005), highlighting the need to address both body image perception and psychological factors in preventive strategies. These findings are consistent with previous research indicating that students with overweight and higher PBF are more likely to adopt maladaptive eating behaviors [50,51] and that a poorer perception of body image is linked to a higher risk of developing EDs [52,53].

In addition to body shape concern, another relevant factor that showed significant associations was PA. A direct relationship was found between PA level and the risk of developing EDs. Women in the higher PA group (Group 1) had a significantly greater risk of developing EDs compared to those in the lower PA group (Group 2). The prevalence of high ED risk was 16.67% in Group 1 versus 2.08% in Group 2, with a prevalence ratio of 8, indicating that the risk of EDs is eight times higher in the more active group. This association may be explained by a heightened concern with physical appearance among those who engage in more exercise, making them more susceptible to adopting disordered eating behaviors. Although the present study did not directly assess exercise motivation, recent studies have shown that appearance-related concerns are a significant driver of physical activity, particularly among university students in health-related fields. For instance, a fear of negative appearance evaluation has been associated with both disordered eating attitudes and increased engagement in physical activity. Likewise, appearance-oriented motivation has been linked to orthorexic tendencies and restrictive eating behaviors in student populations [54,55].

Recent evidence further supports this association. For instance, maladaptive or excessive exercise has been found to be highly prevalent among individuals with EDs and is linked to more severe psychopathology, including depression, anxiety, and obsessive–compulsive traits. Moreover, individuals engaging in such exercise behaviors are more likely to seek treatment, underscoring the clinical relevance of assessing not only the quantity but also the quality and motivation behind PA [56,57]. Taken together, the findings in the present study and previous research [56,57,58] underscore the complexity of the relationship between physical activity and eating disorders, particularly in contexts where exercise becomes a compulsive strategy for weight regulation.

Furthermore, it has been shown that certain psychological aspects related to exercise and/or PA—such as exercise dependence and obsessive–compulsive symptoms—may mediate the relationship between personality traits and EDs [59]. Recent findings suggest that sociocultural pressures and perfectionistic tendencies, particularly the desire to appear effortlessly perfect, are associated with increased body dissatisfaction and pathological eating and exercise behaviors, mediated by self-criticism [60,61]. This is consistent with recent evidence showing that maladaptive perfectionism is a significant predictor of disordered eating symptoms and a reduced quality of life among university students [62].

Regarding self-esteem, a significant inverse relationship was found with body shape concern. Students with higher BSQ scores (indicating greater body dissatisfaction) had lower RSE scores, reflecting lower self-esteem. This result is consistent with previous studies conducted among medical [10] and kinesiology students [63], which also reported a negative association between body image perception and self-esteem.

With respect to body composition variables, although no significant differences were observed in weight or BMI between groups with different levels of PA, significant differences were found in SMM. Students in the higher PA group (Group 1) had significantly greater SMM, suggesting that, while the amount of PA may not directly correlate with these anthropometric indicators, it can positively influence body composition, particularly in terms of muscle gain. This finding aligns with previous research reporting higher SMM in women with more PA [64].

The relationship between self-esteem, body shape concern, and PA level was also explored. No statistically significant differences were found in RSE scores (*p* = 0.32; Cohen’s d = −0.21) or BSQ scores (*p* = 0.44; Cohen’s d = 0.16) between the two groups. However, certain trends were observed: the more active group showed slightly lower self-esteem scores and slightly higher BSQ scores, which may have clinical relevance and warrants further investigation in future studies. These trends are in line with recent post-pandemic findings, which report that university students experienced changes in physical activity patterns and body image perception during and after COVID-19 lockdowns, often accompanied by increased body dissatisfaction and altered eating behaviors [65].

Regarding EDs (*p* = 0.05; Cohen’s d = 0.40; CI: −0.007–0.8) and MEDAS (*p* = 0.07; Cohen’s d = 0.37; CI: −0.03–0.78), although the differences between groups did not reach conventional statistical significance, the observed effect sizes and CI suggest that these may hold clinical relevance. Specifically, students in the more physically active group scored higher in both ED risk and adherence to the Mediterranean diet. These findings may reflect greater nutritional awareness among those engaging in higher levels of physical activity, although they could also indicate heightened concern with body control. Moreover, these results contrast with previous studies that have linked lower levels of physical activity to a greater tendency toward dysfunctional eating behaviors, underscoring the need for further research into the complexity of these relationships [66]

Overall, the results of the present study highlight the need to consider not only the quantity but also the quality of PA, as well as the psychological and social context in which it is practiced. Future interventions should take these factors into account to ensure that the promotion of PA—rightly regarded as a health-promoting behavior—does not inadvertently generate pressure or anxiety related to body image and eating. It is essential to adopt a holistic approach that integrates physical, emotional, and cognitive dimensions. Recent integrative models, such as the Mindfulness–Exercise–Nutrition (MEN) framework, emphasize the importance of combining physical activity, healthy eating—particularly the Mediterranean diet—and mindfulness practices to mitigate the risk of eating disorders and enhance psychological resilience [67]. Similarly, the concept of physical literacy supports a holistic approach by integrating cognitive, affective, and physical dimensions to promote sustainable and health-enhancing physical activity behaviors [68].

Moreover, the early detection of EDs in PA populations—especially among individuals who exhibit compulsive exercise behaviors or motivations focused solely on appearance—is crucial for enabling timely intervention and preventing the chronic progression of these disorders [69]. Preventive strategies should include the training of professionals in the educational, healthcare, and sports sectors to help them to identify warning signs and provide appropriate guidance.

## 5. Conclusions

The results of this study suggest that the level of PA may be related to certain psychological and nutritional factors associated with the risk of developing EDs among female university students in health sciences. Significant differences were observed between those who engage in higher versus lower levels of exercise. Notably, a higher level of PA is not necessarily associated with better body shape concern or higher self-esteem.

Body shape concern emerged as a predictive factor for ED risk, closely related to anthropometric parameters such as weight and PBF, as well as to lower levels of self-esteem. These findings reinforce the importance of addressing physical, psychological, and behavioral aspects in an integrated manner when designing preventive strategies.

This highlights the need to consider the context, intensity, motivations, and emotional consequences of exercise in the prevention and early detection of these disorders. Among the limitations of this study, the heterogeneity of the sample stands out, as it was composed of students from various health-related degree programs. This diversity may have influenced the results due to the differing characteristics and habits of each group. Additionally, the assessment of physical activity levels relied solely on the IPAQ questionnaire, which limits the precision and depth of the analysis.

The findings underscore the importance of promoting PA practices that focus on well-being, enjoyment, and holistic health, rather than on body modification or aesthetic performance. An integrated approach that combines the promotion of PA with nutritional education, body acceptance, and the strengthening of self-esteem will be key to preventing EDs and promoting the overall well-being of female university students.

Regarding future lines of research, it would be advisable to incorporate objective methods for measuring physical activity, such as monitoring devices or accelerometry, to complement self-reported questionnaires and allow for a more accurate assessment of physical behavior. Additionally, conducting longitudinal studies would be valuable to establish causal relationships between physical activity levels, body image perception, and the risk of developing eating disorders. It is also recommended to explore in greater depth the motivations that drive physical activity—whether related to health, aesthetics, or performance—and how these motivations influence students’ mental and nutritional health.

## Figures and Tables

**Figure 1 healthcare-13-01679-f001:**
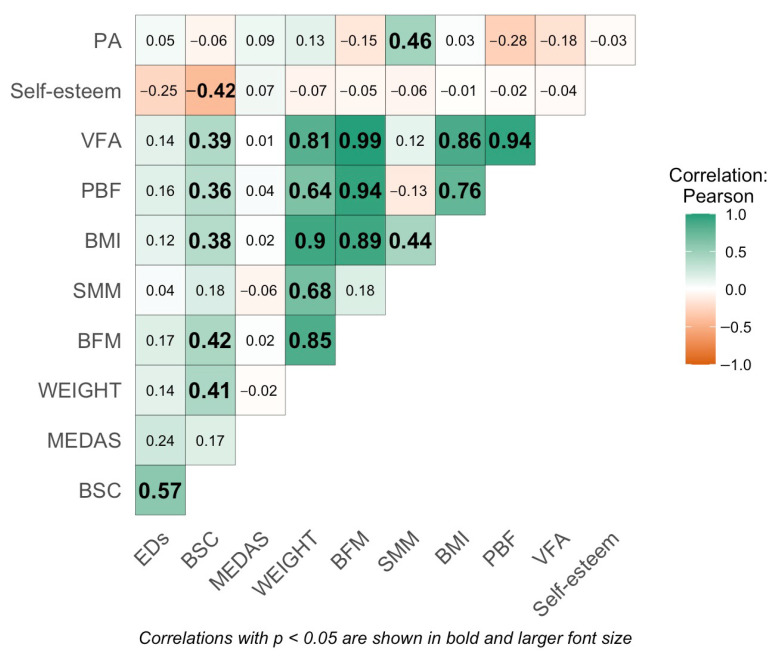
Correlation matrix. Abbreviations: EDs, eating disorders; PA, physical activity; MEDAS, Mediterranean diet adherence; SMM, skeletal muscle mass; BMI, body mass index; BFM, body fat mass; PBF, percent body fat; VFA, visceral fat area; BSC, body shape concern.

**Figure 2 healthcare-13-01679-f002:**
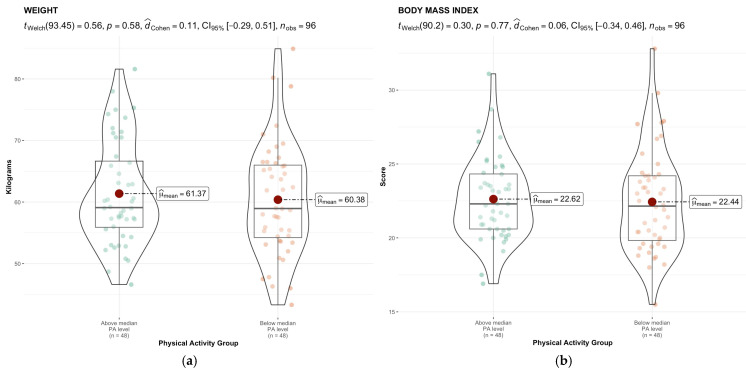
Comparison between groups and central tendency and distribution values of nutritional status: (**a**) Weight; (**b**) BMI; (**c**) SMM; (**d**) PBF; (**e**) BFM; (**f**) VFA.

**Figure 3 healthcare-13-01679-f003:**
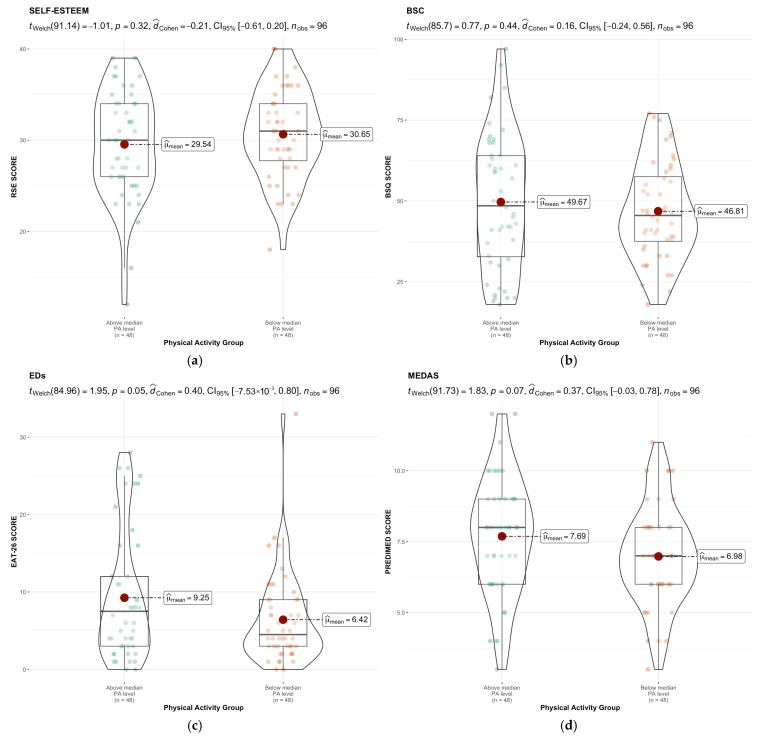
Comparison between groups and central tendency and distribution values of (**a**) Self-esteem; (**b**) BSC; (**c**) EDs; (**d**) MEDAS.

**Figure 4 healthcare-13-01679-f004:**
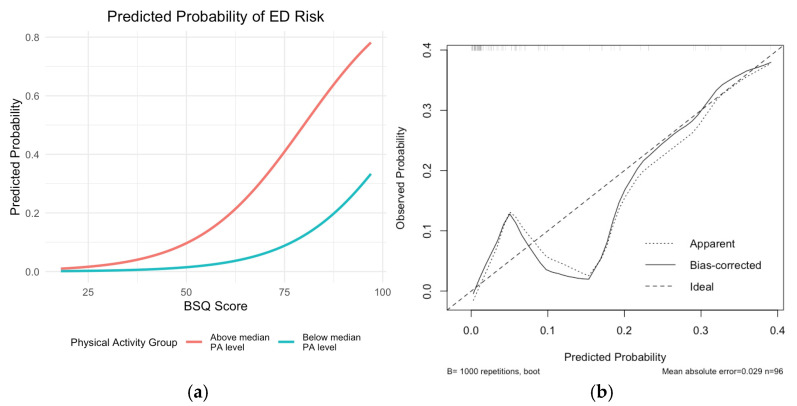
Predicted probability of ED risk (**a**) and calibration curve (**b**).

**Table 1 healthcare-13-01679-t001:** Descriptive statistics (mean and SD in parentheses).

Variable	General	Group 1	Group 2
Participants (n)	96	48	48
Age (years)	20.00 (2.44)	20.58 (1.75)	19.83 (1.21)
Weight (kg)	60.88 (8.63)	61.37 (8.32)	60.38 (8.99)
EDs (EAT26 score) *	7.83 (7.22)	9.25 (8.20)	6.42 (5.84)
PA (METs/week) *	3049.60 (2511.20)	4779.16 (2504.69)	1320.05 (602.48)
MEDAS (PREDIMED score) *	7.33 (1.92)	7.69 (2.04)	6.98 (1.74)
SMM (kg) *	23.36 (2.82)	24.01 (3.03)	22.71 (2.46)
BMI (score) *	22.53 (3.10)	22.62 (2.77)	22.44 (3.42)
BFM (kg) *	18.04 (6.40)	17.52 (6.09)	18.55 (6.73)
PBF (%) *	28.99 (7.05)	28.01 (7.12)	29.97 (6.91)
VFA (cm^2^) *	80.9 (35.4)	77.10 (33.48)	84.75 (37.23)
BSC (BSQ score) *	48.24 (18.03)	49.67 (20.69)	46.81 (15.00)
Self-esteem (RSE score)	30.09 (5.36)	29.54 (5.82)	30.65 (4.87)

* EDs, eating disorders; PA, physical activity; MEDAS, Mediterranean diet adherence; SMM, skeletal muscle mass; BMI, body mass index; BFM, body fat mass; PBF, percent body fat; VFA, visceral fat area; BSC, body shape concern.

**Table 2 healthcare-13-01679-t002:** Prevalence of high risk of EDs based on group membership.

PA Group	Prevalence of No Risk of EDs	Prevalence of Risk of EDs
Group 1 (More actives)	83.33%	16.67%
Group 2 (Less actives)	97.92%	2.08%

**Table 3 healthcare-13-01679-t003:** Logistic model coefficient.

Variable	Coefficient	Standard Error	z-Value	*p*-Value	CI (95%)	
					Lower	Upper
Intercept	−5.965	1.768	−3.374	0.001	−10.156	−3.047
PA Group	−1.968	1.120	−1.756	0.079	−4.948	−0.095
BSC	0.075	0.027	2.804	0.005	0.029	0.137

Note: Reference = Group 2, Contrast = Group 1 (more active group with higher risk).

## Data Availability

Data are available upon request from the corresponding author.

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
