# Peer review of "Physical Activity and Psychonutritional Correlates of Eating Disorder Risk in Female Health Science Students"

_healthcare, 2025, doi:10.3390/healthcare13141679_

Round 1
Reviewer 1 Report
Comments and Suggestions for Authors
Dear Authors,
This manuscript investigates the relationship between physical activity (PA), nutritional status, self-esteem, and body image, and their influence on the risk of eating disorders (EDs) in female university students enrolled in health-related academic programs. The research is timely and addresses a public health issue of increasing relevance. The study design is well-structured, with appropriate methodology, validated instruments, and the use of modern statistical tools. However, additional methodological clarification and references to the most recent literature are required.
Keywords: Please avoid repeating terms already included in the title. Consider using MeSH (Medical Subject Headings) terms, as found at https://www.ncbi.nlm.nih.gov/mesh/.
Introduction: The introduction would benefit from mentioning specific sports disciplines, supported by relevant references, that, when combined with the Mediterranean diet, have been shown to positively influence body composition and self-esteem. Conversely, it is well-established that certain disciplines are more likely than others to be associated with the development of eating disorders. Including this information would enhance the specificity and originality of the manuscript.
Table 1: It is apparent that the values represent means and the values in parentheses represent standard deviations. However, this should be explicitly stated in the table caption for clarity.
Figure 1: The "X" denoting non-significance overlaps with the numbers inside the boxes, which impairs readability. Please adjust the figure layout to move the "X" outside the numbers or consider using a different notation.
Additionally, the figure lacks precise correlation coefficients, which would improve visual interpretation. The phrase “not significant at p < 0.05” may cause confusion. Do the numbers marked with an "X" correspond to p-values > 0.05? Please clarify this point and revise the figure caption accordingly for greater clarity.
Statistical Analysis: The choice of statistical tools is modern and appropriate. However, key assumptions for logistic regression, such as checks for multicollinearity and model calibration, are not adequately addressed. Please include diagnostic tests and residual analyses to assess the validity of the regression models.
Furthermore, p-values that approach statistical significance are overemphasized in the discussion without sufficient correction for multiple comparisons. Avoid interpreting non-significant trends as meaningful unless supported by strong effect sizes.
Methods: The methods section is well-detailed and adheres to STROBE guidelines. The tools used (e.g., IPAQ, EAT-26, PREDIMED, BSQ, RSE) are appropriately validated and referenced. However, the classification threshold for physical activity (2373 METs/week) is based on the sample median and lacks reference to established health guidelines (e.g., WHO or ACSM standards). This limits comparability across studies. Please justify this cutoff or reanalyze the data using standardized PA categories to enhance generalizability.
References and Discussion: The manuscript includes relevant literature on EDs, physical activity, and adherence to the Mediterranean diet. However, at lines 385–386, the statement “It is essential to adopt a holistic approach that integrates physical, emotional, and cognitive dimensions” lacks a supporting citation. Please refer to recent literature to strengthen this point.
For example, Baldassano et al. (2023) present an integrative model—Mindfulness–Exercise–Nutrition (MEN)—that highlights how physical activity and diet, particularly the Mediterranean diet, can mitigate eating disorder and enhance psychological resilience. They also propose a practical framework for intervention. Citing this work, or similar studies, would strengthen the theoretical foundation of your manuscript and enhance its practical implications:
Baldassano et al. (2023). Fighting the Consequences of the COVID-19 Pandemic:
Mindfulness, Exercise, and Nutrition Practices to Reduce Eating Disorders and Promote Sustainability. Sustainability, 15(3), 2120. https://doi.org/10.3390/su15032120
Author Response
We sincerely appreciate your detailed and constructive feedback on our manuscript. We especially value your recognition of the study’s relevance, methodological rigor, and the use of validated instruments alongside current statistical approaches. We have carefully reviewed each of your observations and made the corresponding revisions to the manuscript. Below, we provide a point-by-point response outlining the changes made and the rationale behind them. We hope that these modifications have successfully addressed your comments and contributed to improving the overall quality and clarity of the manuscript.
Comment 1:
Keywords: Please avoid repeating terms already included in the title. Consider using MeSH (Medical Subject Headings) terms, as found at https://www.ncbi.nlm.nih.gov/mesh/.
Respond 1:
Following your recommendation, we have revised the keyword section to avoid repetition of terms already present in the title and to better align with standardized terminology using MeSH (Medical Subject Headings) (Line 32)
Comment 2:
The introduction would benefit from mentioning specific sports disciplines, supported by relevant references, that, when combined with the Mediterranean diet, have been shown to positively influence body composition and self-esteem. Conversely, it is well-established that certain disciplines are more likely than others to be associated with the development of eating disorders. Including this information would enhance the specificity and originality of the manuscript.
Respond 2:
Thank you for your constructive suggestion. In response, we have expanded the Introduction to address sport-specific contexts, citing current evidence (Lines 52-64).
Comment 3:
Table 1: It is apparent that the values represent means and the values in parentheses represent standard deviations. However, this should be explicitly stated in the table caption for clarity.
Respond 3:
Thank you for your observation. The requested information has been included in the manuscript and is now explicitly stated in the caption of Table 1 for clarity (Lines 300-302)
Comment 4:
Figure 1: The "X" denoting non-significance overlaps with the numbers inside the boxes, which impairs readability. Please adjust the figure layout to move the "X" outside the numbers or consider using a different notation.
Additionally, the figure lacks precise correlation coefficients, which would improve visual interpretation. The phrase “not significant at p < 0.05” may cause confusion. Do the numbers marked with an "X" correspond to p-values > 0.05? Please clarify this point and revise the figure caption accordingly for greater clarity.
Respond 4:
Thank you for your observation. Figure 1 has been revised to improve readability (Lines 313-314)
Comment 5:
Statistical Analysis: The choice of statistical tools is modern and appropriate. However, key assumptions for logistic regression, such as checks for multicollinearity and model calibration, are not adequately addressed. Please include diagnostic tests and residual analyses to assess the validity of the regression models.
Furthermore, p-values that approach statistical significance are overemphasized in the discussion without sufficient correction for multiple comparisons. Avoid interpreting non-significant trends as meaningful unless supported by strong effect sizes.
Respond 5:
Thank you for your valuable feedback. We have addressed the requested changes by incorporating diagnostic tests for multicollinearity and model calibration into the statistical analysis (lines 270-285) and results (Lines 362-380). Additionally, we have revised the discussion section to avoid overinterpretation of non-significant trends and to clarify the interpretation of p-values, particularly in the context of multiple comparisons (Lines 459-468).
Comment 6:
Methods: The methods section is well-detailed and adheres to STROBE guidelines. The tools used (e.g., IPAQ, EAT-26, PREDIMED, BSQ, RSE) are appropriately validated and referenced. However, the classification threshold for physical activity (2373 METs/week) is based on the sample median and lacks reference to established health guidelines (e.g., WHO or ACSM standards). This limits comparability across studies. Please justify this cutoff or reanalyze the data using standardized PA categories to enhance generalizability.
Respond 6:
Thank you for your thoughtful comment. Due to the relatively high physical activity levels observed in our sample (mean = 3,050 MET-min/week)—likely reflecting the characteristics of an active female population—the majority of participants exceeded established thresholds for moderate or high activity. Specifically, only a small subset of participants fell below 1,500 MET-min/week (n = 28), and even fewer fell below the WHO-recommended minimum of 600 MET-min/week (n = 8). To avoid imbalanced group sizes and sparse data in logistic regression modeling, we opted for a data-driven cutoff based on the sample median (2,373 MET-min/week) to dichotomize physical activity levels. This approach allowed for balanced groups and more stable model estimation, while still capturing meaningful variation in physical activity across participants.
Comment 7:
References and Discussion: The manuscript includes relevant literature on EDs, physical activity, and adherence to the Mediterranean diet. However, at lines 385–386, the statement “It is essential to adopt a holistic approach that integrates physical, emotional, and cognitive dimensions” lacks a supporting citation. Please refer to recent literature to strengthen this point.
For example, Baldassano et al. (2023) present an integrative model—Mindfulness–Exercise–Nutrition (MEN)—that highlights how physical activity and diet, particularly the Mediterranean diet, can mitigate eating disorder and enhance psychological resilience. They also propose a practical framework for intervention. Citing this work, or similar studies, would strengthen the theoretical foundation of your manuscript and enhance its practical implications:
Baldassano et al. (2023). Fighting the Consequences of the COVID-19 Pandemic: Mindfulness, Exercise, and Nutrition Practices to Reduce Eating Disorders and Promote Sustainability. Sustainability, 15(3), 2120. https://doi.org/10.3390/su15032120
Respond 7:
Thank you for your suggestion. We have incorporated the recommended literature into the discussion section (lines 475-480).
Reviewer 2 Report
Comments and Suggestions for Authors
General Evaluation
The manuscript is interesting and timely exploring the relationship between physical activity (PA) levels and psychological and nutritional variables associated with the risk of eating disorders (EDs) in female university students from Health Sciences programs. The topic is highly relevant, and the study aims to address important issues regarding the complex interplay between PA, body image, and eating behavior. However several methodological and interpretative concerns that should be addressed to strengthen the validity and impact of the manuscript.
Introduction
The introduction is generally well structured and presents a solid synthesis of the relevant literature. However, it is important to note that 46.5% of the cited references are from the last five years, which may undermine the manuscript’s currency. Updating the literature with more recent studies would enhance the relevance and credibility of the theoretical framework.
Methods
The methodology, tools, and procedures are clearly described. Nevertheless, there are significant methodological limitations that compromise the internal validity and interpretation of the findings:
Sample Heterogeneity:
The sample includes students from four distinct degree programs (Sport Sciences: n=9, Nutrition: n=11, Nursing: n=52, Kinesiology: n=20), introducing a high degree of heterogeneity. This is particularly problematic in relation to baseline PA levels, which may differ structurally across disciplines due to curriculum and cultural norms. These differences represent potential confounding factors that are not adequately addressed or controlled in the analysis.
Subjective and Indirect Assessment of Physical Activity:
The use of the International Physical Activity Questionnaire (IPAQ) as the sole instrument for assessing PA presents clear limitations. As a self-report measure, IPAQ is prone to recall bias and social desirability bias, particularly in health-related student populations who may overestimate their PA levels. This undermines the reliability of the PA data used in the analysis.
Results
The results are clearly presented using appropriate tables and figures. The data presentation is well-organized, and the interpretation of the findings is generally coherent. However, the analysis could benefit from addressing potential biases resulting from the heterogeneity of the sample.
Discussion
The discussion provides a comparison with existing literature and draws relevant links to prior findings. Nonetheless, it suffers from limited incorporation of recent studies, with few references from the last five years. Additionally, the interpretation of results should be more closely aligned with the methodological limitations noted above. In particular, the manuscript draws conclusions about appearance-driven motivations for PA, despite not having measured motivation directly.
Conclusions
The conclusions are generally consistent with the study objectives and findings. However, several issues remain:
-
Study limitations and strengths are not clearly acknowledged, which is essential for transparency and scientific rigor.
-
The claim that higher levels of PA may be linked to an increased risk of EDs due to appearance-related motivations is not empirically supported, as motivational constructs were not assessed in the study. This results in speculative interpretations that should be either removed or substantially qualified.
Despite these issues, the authors do attempt to explore the implications of their findings and offer suggestions for future research. This is a valuable contribution and should be further expanded.
Author Response
We sincerely thank you for your thoughtful comments and constructive suggestions. We appreciate your recognition of the relevance and timeliness of our study, as well as your engagement with the methodological and interpretative aspects of the manuscript.
Below, we provide a point-by-point response to each of your comments and outline the corresponding revisions made to strengthen the validity and impact of the manuscript.
Comment 1:
The introduction is generally well structured and presents a solid synthesis of the relevant literature. However, it is important to note that 46.5% of the references cited are from the last five years, which may undermine the manuscript’s currency. Updating the literature with more recent studies would enhance the relevance and credibility of the theoretical framework.
Response 1:
Thank you for your valuable feedback regarding the recency of our references. We have reviewed and updated the Introduction with several recent studies (from the last five years) that reinforce the theoretical framework.
Comment 2:
The methodology, tools, and procedures are clearly described. Nevertheless, there are significant methodological limitations that compromise the internal validity and interpretation of the findings:
Sample heterogeneity: The sample includes students from four distinct degree programs (Sport Sciences: n=9, Nutrition: n=11, Nursing: n=52, Kinesiology: n=20), introducing a high degree of heterogeneity. This is particularly problematic in relation to baseline PA levels, which may differ structurally across disciplines due to curriculum and cultural norms. These differences represent potential confounding factors that are not adequately addressed or controlled in the analysis.
Response 2:
Thank you for your thoughtful comment. In response to this concern, we examined the potential impact of academic program heterogeneity on the analysis as follows: Aware of the potential heterogeneity introduced by including students from different academic programs, we explored the inclusion of “degree program” (DEGREE) as a covariate in the logistic regression model. Although physical activity levels differed significantly across degrees, DEGREE was not a significant predictor of eating disorder risk (p = 0.18), and its inclusion did not substantially improve model fit (AIC decreased only marginally from 47.161 to 47.137). Therefore, in favor of model parsimony and interpretability, we opted to exclude DEGREE from the final model while acknowledging sample heterogeneity as a limitation. (line 355-361)
Comment 3:
Subjective and Indirect Assessment of Physical Activity:
The use of the International Physical Activity Questionnaire (IPAQ) as the sole instrument for assessing PA presents clear limitations. As a self-report measure, IPAQ is prone to recall bias and social desirability bias, particularly in health-related student populations who may overestimate their PA levels. This undermines the reliability of the PA data used in the analysis.
Response 3:
Thank you for highlighting this important point. We acknowledge the limitations associated with the use of the International Physical Activity Questionnaire (IPAQ), particularly the potential for recall bias and social desirability bias, which may be more pronounced in health-related student populations. Despite these limitations, the IPAQ is a widely validated and internationally accepted instrument for assessing physical activity in large-scale studies, including among university students. Its use was selected to ensure comparability with prior research and feasibility within the broader scope of our survey protocol. Nevertheless, we have now explicitly discussed this limitation in the manuscript and recommend that future studies incorporate objective measures of physical activity (e.g., accelerometry) when feasible, to strengthen the reliability of PA assessments (Lines 510-518)
Comment 4:
The results are clearly presented using appropriate tables and figures. The data presentation is well-organized, and the interpretation of the findings is generally coherent. However, the analysis could benefit from addressing potential biases resulting from the heterogeneity of the sample.
Response 4:
We appreciate the reviewer’s comment. As addressed above, we analyzed the influence of academic program (DEGREE) on physical activity and tested its inclusion in the regression model. Although DEGREE was not a significant predictor, we have acknowledged sample heterogeneity as a limitation and discussed its potential impact in the revised manuscript. (Lines 355-361)
Comment 5:
The discussion provides a comparison with existing literature and draws relevant links to prior findings. Nonetheless, it suffers from limited incorporation of recent studies, with few references from the last five years. Additionally, the interpretation of results should be more closely aligned with the methodological limitations noted above. In particular, the manuscript draws conclusions about appearance-driven motivations for PA, despite not having measured motivation directly.
Response 5:
Thank you for your suggestion. To address your comments, we have added new information (lines 414-420; 432-437 and 455-458).
Comment 6:
The conclusions are generally consistent with the study objectives and findings. However, several issues remain:
- Study limitations and strengths are not clearly acknowledged, which is essential for transparency and scientific rigor.
- The claim that higher levels of PA may be linked to an increased risk of EDs due to appearance-related motivations is not empirically supported, as motivational constructs were not assessed in the study. This results in speculative interpretations that should be either removed or substantially qualified.
Despite these issues, the authors do attempt to explore the implications of their findings and offer suggestions for future research. This is a valuable contribution and should be further expanded.
Response 6:
We are thankful for your detailed evaluation. Regarding the reviewer’s comment that the statement linking higher levels of physical activity (PA) to an increased risk of eating disorders (EDs) due to appearance-related motivations is not empirically supported, we have considered this observation and decided to remove the corresponding paragraphs. In response to the suggestion to include methodological limitations, these have been addressed in the paragraphs located at lines 501-505 of the revised manuscript. Additionally, as proposed, we have incorporated paragraphs at lines 510-518 discussing the implications of the findings and outlining future research directions.
Reviewer 3 Report
Comments and Suggestions for Authors
Introduction:
- Authors provided sufficient literatures to explain the correlation about the PA engagement and body image.
- However, the linkage between "adherence to the Mediterranean diet" to body dissatisfaction and the risk of developing eating disorders is being forcefully impose in the introduction (line59 to 68). Why only "adherence to the Mediterranean diet" but not other special diets being included in this study? Authors need to provide stronger rationale for their choice.
Materials and Methods:
- Clearly written, especially for the section related to sample size determination.
- For 2.3 procedure (line 113-119), better provide reasons for administering the questionnaires for 4 separate weeks. What is the reasons for administering the questionnaires in this specific sequence?
- Nutritional status was assessed by anthropometric parameters. It is acceptable according to the ABCD of nutritional assessment. However, readers may not know about it and they may consider those assessment items as anthropometric measures rather than data related to nutritional status. Authors can add a few words to mention about it.
Results:
- Fairly presented
Discussion:
1. More literatures should be provided in discussing the results of this study so that readers can obtain a more holistic pictures of this topic. For example, add some findings or discussion of previous studies about the association between exercise and eating disorders (line 344 to 349).
Author Response
Thank you very much for your thoughtful and constructive feedback. We sincerely appreciate your recognition of the strengths of our manuscript, particularly the clarity of the methodology and the relevance of the literature supporting the relationship between physical activity and body image.
Your observations regarding the conceptual framing of the Mediterranean diet, the procedural details of questionnaire administration, and the interpretation of anthropometric data have been extremely helpful in refining key aspects of our work.
Below, we provide a point-by-point response to each of your comments, along with a summary of the corresponding revisions made to strengthen the manuscript.
Comment 1:
- Authors provided sufficient literatures to explain the correlation about the PA engagement and body image.
- However, the linkage between "adherence to the Mediterranean diet" to body dissatisfaction and the risk of developing eating disorders is being forcefully impose in the introduction (line59 to 68). Why only "adherence to the Mediterranean diet" but not other special diets being included in this study? Authors need to provide stronger rationale for their choice.
Response 1:
Thank you for your comment. We chose to focus on the Mediterranean diet because emerging evidence indicates that Mediterranean diet adherence is inversely associated with eating disorder risk and body dissatisfaction—more clearly than many other dietary patterns.
While other diets have been studied, none show similarly consistent protective effects against core eating disorder symptoms or incidence. Given the consistency, dose–response relationship, and longitudinal design of Mediterranean diet specific studies—and the sociocultural relevance of the Mediterranean diet in our Spanish/Female Health Science student sample—we consider Mediterranean diet adherence to be the most appropriate dietary focus for this study.
Comment 2:
- Clearly written, especially for the section related to sample size determination.
- For 2.3 procedure (line 113-119), better provide reasons for administering the questionnaires for 4 separate weeks. What is the reasons for administering the questionnaires in this specific sequence?
Response 2:
We appreciate the reviewer’s suggestion. As requested, we have clarified in the manuscript the rationale for distributing the questionnaires over four consecutive weeks. This approach was intended to minimize response bias and improve data quality by allowing participants to respond with greater attention and accuracy. The revised explanation appears in lines 138–141 of the manuscript.
Comentario 3:
- Nutritional status was assessed by anthropometric parameters. It is acceptable according to the ABCD of nutritional assessment. However, readers may not know about it and they may consider those assessment items as anthropometric measures rather than data related to nutritional status. Authors can add a few words to mention about it.
Response 3:
We appreciate the reviewer’s observation. To clarify that the anthropometric parameters used in this study are a standard part of nutritional assessment, we have added a reference to the ABCD framework (Anthropometric, Biochemical, Clinical, and Dietary) in the relevant section of the manuscript. This helps ensure that readers unfamiliar with the classification understand that the anthropometric data presented were used to evaluate nutritional status (Lines 216-219).
Comment 4:
- More literatures should be provided in discussing the results of this study so that readers can obtain a more holistic pictures of this topic. For example, add some findings or discussion of previous studies about the association between exercise and eating disorders (line 344 to 349).
Response 4:
We have incorporated new studies as you suggested and added their findings, as shown in lines 421 to 429.
Round 2
Reviewer 1 Report
Comments and Suggestions for Authors
Dear Authors,
I thank you for your revisions. The manuscript has shown a clear improvement in both scientific and methodological rigor, particularly through the inclusion of more detailed information on the statistical analyses. You have taken care of all the comments clearly and satisfactorily. I have no further remarks.
Best regards,
the reviewer
Author Response
We thank the reviewer for their thorough assessment and thoughtful remarks. We are pleased that the improvements in the methodological and statistical reporting were satisfactory and that all previous concerns were adequately addressed. We appreciate their time and contribution to enhancing the quality of this manuscript.